# Micronutrient Analysis of Gluten-Free Products: Their Low Content Is Not Involved in Gluten-Free Diet Imbalance in a Cohort of Celiac Children and Adolescent

**DOI:** 10.3390/foods8080321

**Published:** 2019-08-07

**Authors:** Idoia Larretxi, Itziar Txurruka, Virginia Navarro, Arrate Lasa, María Ángeles Bustamante, María del Pilar Fernández-Gil, Edurne Simón, Jonatan Miranda

**Affiliations:** 1Gluten Analysis Laboratory of the University of the Basque Country, Department of Nutrition and Food Science, University of the Basque Country, UPV/EHU, 01006 Vitoria, Spain; 2GLUTEN3S research group, Department of Nutrition and Food Science, University of the Basque Country, UPV/EHU, 01006 Vitoria, Spain

**Keywords:** celiac disease, gluten-free diet, gluten-free product, micronutrient, vitamin and minerals, dietary recommendation

## Abstract

Data about the nutritional composition of gluten-free products (GFP) are still limited. Most studies are based on ingredient and nutrition information described on the food label. However, analytical determination is considered the gold standard for compositional analysis of food. Micronutrient analytical content differences were observed in a selection of GF breads, flakes and pasta, when compared with their respective gluten-containing counterparts. In general terms, lower iron, piridoxin, riboflavin, thiamin, niacin, folate, manganese and vitamin B5 can be underlined. Variations in biotin and vitamin E content differed among groups. In order to clarify the potential contribution of the GFP to the gluten-free diet’s (GFD) micronutrient shortages, analytical data were used to evaluate GFD in a cohort of celiac children and adolescent. Participants did not reach recommendations for vitamin A, vitamin E, folic acid, vitamin D, biotin, iodine, and copper. It does not seem that the lower micronutrient content of the analyzed GFP groups contributed to the micronutrient deficits detected in GFD in this cohort, whose diet was not balanced. Nevertheless, GFP fortification for folate and biotin is proposed to prevent the deficiencies observed in GFD, at least in the case of pediatric celiac disease.

## 1. Introduction

Celiac disease (CD) is a chronic immune-mediated inflammatory pathology triggered by the gluten in the diet of genetically predisposed individuals. The need to avoid this protein in the diet of celiac people brought about some years ago the development of specific cereal-based gluten-free products (GFP). Despite the fact that these GFP allowed them to include a wide variety of foods in their diets, in recent years researchers have highlighted differences in the nutrient composition of GFP with respect to gluten containing counterparts [1,2], leading to a minor health rating in some food-groups [3,4].

It is important to note that most of the studies about the nutrient composition of the GFP are based on ingredients and the nutrition information described on the food label [2,3,4]. To improve these data, some works, such as that carried out by Mazzeo et al. (2015) [5], take advantage of the retention factors for each nutrient, including losses due to heating or other food preparation steps. However, analytical determination is considered the gold standard for composition analysis of food. Accurate analysis could also provide detailed information about vitamins and minerals, which is not totally or commonly available on label [6]. Therefore, access to micronutrient data is already restricted to hardly any research [7,8,9].

Furthermore, a gluten-free diet (GFD) often implies some nutritional imbalances, as recognized in the literature [10,11]. Not only have inadequate fat, protein, sugar and fiber consumption been observed in GFD, but also a poor intake of micronutrients such as iron, zinc, magnesium, calcium, folate, vitamin D and B_12_ [12]. Similarly, celiac people seem to have lower blood values for hemoglobin, ferritin, vitamin D, and copper than the rest of the population [13,14]. There has been speculation about whether the characteristic composition of GFP is responsible for GFD inadequacy. A potential correlation between both facts has been proposed by others [15].

In the case of GFP, the use of raw material such as unenriched rice or maize refined flours, gums or enzymes in their formulation could lead to a different composition compared to their gluten containing homologues [16]. Moreover, as the micronutrient content of gluten-free pseudocereals and legumes is higher than that of the gluten free cereals [15,17], some authors proposed to promote their use in GFP formulation [12,18,19].

In view of the above, the aim of this study was to assess analytically the macronutrient and micronutrient content of a selection of GF breads, flakes and pasta, and to compare it with their respective gluten-containing counterparts. Additionally, in order to clarify the potential contribution of the GFP to the GFD’s micronutrient shortages, vitamin and mineral analytical data were used to evaluate GFD in a cohort of celiac children and adolescents.

## 2. Materials and Methods

### 2.1. Analytical Nutrient Content of GF Bread, Breakfast Cereals and Pasta

The measured samples were thirty-seven selected GFP signed with the Crossed Grain symbol: 13 breakfast cereals, 12 breads and 12 pasta products (Appendix A). All the food items were purchased from the local market (Vitoria, Spain) and they were stored frozen (−20 °C) until analyzed. The analytically determined composition of GF foodstuffs was compared with the data of equivalent gluten-containing breads (*n* = 19), breakfast cereals (*n* = 18), and pasta products (*n* = 8), analyzed in the same way and at the same time for macronutrients, and with micronutrient data obtained from the Spanish Food Composition Database—BEDCA database [20]. These results were also compared with the data described in the food label of GFPs.

Analysis of the nutrient content of foodstuff has been carried out using official methods. Crude protein content was determined by the Kjeldahl method (AOAC, 960.52A) [21] in a Foss Kjeltec™ distillation unit (Höganäs, Sweden). Fat content was analyzed by the Soxhlet extraction method based on the official method (AOAC, 2003.05) [21], using a Soxtherm extraction system (Gerhardt, Bonn, Germany). Determinations were performed in duplicate.

For mineral determination, microwave-assisted digestion was carried out in a closed microwave device Mars 5 (CEM, Vertex, Barcelona, Spain) equipped with 8–24 teflon vessels and temperature controllers. The quantitative analysis of selenium, manganese and cooper was performed by using ICP-MS (7700x, Agilent Technologies, Palo Alto, CA, USA) and MicroMist micro-uptake glass concentric nebulizer (Glass Expansion, West Melbourne, Victoria, Australia). ICP-OES (Horiba Jobin Yvon Activa, Kyoto, Japan) was used with a quartz Meinhard concentric nebulizer, a Scott-type spray chamber and a standard quartz sheath connection between the spray chamber and the torch in the case of calcium, sodium, zinc and iron quantification. Working standard solutions of Ca, Na (0–20 mg/L), Fe, Zn and Se (0–100 µg/L) were prepared immediately prior to their use, by stepwise dilution of certified standard multi-element solution (100 mg/L) (Merck, Darmstadt, Germany) with HNO3 1.0 % *v*/*v* (Merck, Darmstadt, Germany). Additionally, a 10 mg/L multi-element standard solution (Y, Rh) from Inorganic Ventures (Equilab, Madrid, Spain) was also used as the internal standard in direct ICP-MS analysis.

As a step prior to vitamin quantification, samples were extracted by liquid-liquid extraction using an aqueous acidic mixture, centrifuged and filtrated, except for vitamin E. Biotin, Folate, Niacin, Pyridoxine, Riboflavin, Thiamine, vitamin B_5_ and B_12_ were measured by liquid chromatography (LC) with triple quadrupole mass spectrometry detection. High purity (>95%) standards (Merck, Darmstadt, Germany) were used for the identification of each vitamin by positive ionization of the electrospray and multiple reaction monitoring. Quantification was developed using the standard addition method. Vitamin E determination was carried out by previous saponification of the samples, followed by a liquid-liquid extraction and purification of the extracts. Afterwards, high performance LC with the fluorescence detector method was used to analyze vitamin E in each extract. Quantification was performed by an external calibration method using the calibration curve of the tocopherol standard (Merck, Darmstadt, Germany). Analytical determinations of micronutrients were carried out once in each sample, but it was verified before the analysis that the reproducibility of the methods was less than 5%.

As mentioned, the micronutrient content of GF foodstuffs was compared with that of gluten-containing counterparts, obtained from the Spanish Food Composition Database—BEDCA database- [20]. Data for biotin in all studied food groups and copper in cereals were obtained from McCance and Widdowson’s “composition of foods integrated dataset” from the United Kingdom [22]. No available data were found with regard to the manganese content of cereal flakes in food composition databases from the UK, Australia, the USA or Spain [20,22,23,24].

### 2.2. Dietary Assessment: Participants and Procedure

Eighty-three minor celiac (age: 3 to 18 years; 53 girls and 30 boys) from the Basque Country took part in the study. The age of the participants was selected due to their higher consumption of GFP compared to adults [25,26]. All participants received oral and written information about the nature and purpose of the survey, and all of them gave written consent for involvement in the study. This study was approved by the Ethical Committee in University of Basque Country (CEISH/76/2011 and CEISH/194M/2013).

The dietary assessment followed in the research was described elsewhere [26]: three days food records (two weekdays and one weekend day) were selected for each patient, 24-h food recalls (24HRs) were filled in by each celiac patient. Micronutrient intake was calculated by a computerized nutrition program system (AyS, Software, Tandem Innova, Inc., Huesca, Spain). The analytically measured vitamin and mineral content of tested GF products was added into the food composition database of the program before calculations. Dietary reference intakes (DRI) for the Spanish population issued by the Spanish Societies of Nutrition, Feeding and Dietetics (FESNAD) in 2010 were taken as references for the interpretation of the 24HRs [27].

### 2.3. Statistical Analysis

Results are presented as mean ± standard deviation (SD) of the mean. Statistical analysis was performed using SPSS 24.0 (SPSS Inc., Chicago, IL, USA). After confirming the normal distribution of lipid and protein content variables using Shapiro-Wilks normality, paired-samples student’s t test was used for comparison. Due to their skewed distribution, micronutrients variables for analytical and database information were analyzed by Mann–Whitney *U*. The level of significance was set to *p* < 0.05.

## 3. Results and Discussion

### 3.1. Macronutrient Content of GF Rendered Foods

With the aim of assessing representative products of a GFD, GFP from the three main cereal food-types contributing to a balanced diet, such as flakes, pasta and bread, were selected. Protein and lipid contents of the three GFP groups analyzed are shown in Table 1. Results were compared to the nutritional composition of their gluten-containing counterparts. With regard to breads, lipid content was higher and the protein content was lower than that of gluten containing products. Similarly, GF bread has been described as poor in proteins and rich in fat content by others [28]. GF pasta provided a lower protein amount, although the comparison to gluten containing pasta did not reach statistical significance. In general terms, lower protein content in GFP than in their counterpart has been proposed by previous research [2,3,4]. Nevertheless, and in good accordance with our data, Missbach et al. did not observe this pattern in flakes [2].

Some clues for justifying the results could be extracted from the list of ingredients of GFP (Appendix A). Rice and maize flours are extensively used in GFP, especially in breads, and according to composition databases, their protein content is lower than that of wheat. Moreover, maize and rice starches, usually added as a substitute, are especially poor in this macronutrient. For pasta and flakes, other ingredients could hinder the protein deficit, such as cocoa or eggs, soy protein or meat from the filled pasta. For lipids, the use of additives like mono and diglycerides of fatty acids (E-471) in GFP, especially in breads, could affect the final composition. However, this study did not consider the label information of ingredients of GCP, thus making conclusive statements is not possible.

It is important to point out that the comparative study between GFP and their homologues with gluten in the present work was performed as suggested by Staudacher and Gibson [6], by direct analytical methods and in paired form. As stated in the introduction, most of the studies evaluating the differences between both foodstuffs are based on nutrition information taken from the food label. For this reason, the analytical results obtained were compared to those reported in the nutritional panel information and some interesting data were collected. With regard to bread, experimental data reported a lower lipid (23%, *p* = 0.07) and higher protein (37%; *p* = 0.03) content than that supplied by the label. Similarly, in the case of cereal flakes, the measured protein amount was higher (19%; *p* = 0.04). No differences were observed between analyzed and labelled data in GF pasta.

In view of Regulation (EU) No 1169/2011 [29], the declared values on labels shall be average values based on (a) the manufacturer’s analysis of the food; (b) a calculation from the known or actual average values of the ingredients used; (c) a calculation from generally established and accepted data. It is not possible for us to determine how each manufacturer calculated label information. However, it must be highlighted that nutrient variations observed in bread types are not within the tolerance ranges between label information and our direct food analysis (tolerance ranges: ±1.5 g for lipids and ±2 g for proteins, when its content in food is <10 g per 100 g). This information brings to light that previous studies about bread described in the literature could be reconsidered, and additionally, it validates, in part, others about pasta and cereals.

### 3.2. Micronutrient Content of GFP, Compared to Gluten-Containing Products

Despite the growing market of the GFP [30], data about their vitamin and mineral contribution remain scarce. Moreover, the data found in the literature are usually calculated from ingredients and their composition databases, which has been proposed to lead to overestimation [5]. Table 2 shows analytical micronutrient content of GF bread, flakes and pasta, compared to that of their gluten-containing counterparts. Lower iron, piridoxin, riboflavin and thiamin content was found in the three GFP groups analyzed. Niacin reduction was observed in GF flakes and breads. With regard to iron, similar results were found by Rybicka [8], who described that 273 of 408 GFP analyzed fulfilled less than 10% of recommended nutrient intake per portion and only 23 products were major contributors to daily intake (over 25% of recommendation intake per portion). In a study performed with 368 GFP, including flours, breads, pasta and cold cereals, overall it was observed that these kinds of products contained lower amounts of thiamin, riboflavin and niacin than the wheat product they were intended to replace [31]. These results are in line with the results obtained in the present study.

Folate content was lower in GF flakes and pasta types; manganese amount was lower only in GF pasta, and that of vitamin B_5_ in GF flakes. As stated before, commonly used ingredients for GFP are maize and rice flours as well as a variety of starches (potato, corn), among others. It seems that removal of protein-rich fractions from flours may result in dramatic depletion of folates. Additionally, rice flours are not very rich in this vitamin [9]. In fact, we calculated a reduction of almost 80% of folate content in rice flour with respect to wheat flour (*p* = 0.05) comparing the nutrient composition of both flours obtained from food composition databases from the UK, Australia, the USA or Spain [20,22,23,24].

Several studies have claimed lower zinc and copper and higher sodium content for GFP [4,32]. However, no significant differences in those minerals were found in our data.

Finally, biotin content differed widely among groups, being higher in cereal flakes and lower in pasta GFP than in their counterparts. Moreover, we found that some GF cereals were fortified with biotin, thus explaining its higher content in this GF food group. Similarly, although vitamin E contribution from GFP was lower in flakes, no differences were observed in pasta and bread. Moreover, it is worth mentioning that half of the analyzed bread types showed a formulation with sunflower oil (Appendix A), which led to higher vitamin E content in those specific stuffs.

It is important to point out that food technology interventions to improve the shelf life and rheological properties of GFP have influenced their nutritional profile [12]. In order to avoid the absence of the mentioned micronutrients without fortifying foodstuffs, different strategies can be proposed: avoiding starch as a major ingredient, sourdough fermentation, and using less popular grain GF flour such as that from pseudocereals (buckwheat, quinoa, amaranth and teff) or legumes, including wholemeal forms of gluten-free cereals [18,19,33,34]. In our samples, only one out of twelve foodstuffs analyzed in each group contained pseudocereals in their ingredients list (4 to 5 g in 100 g), reflecting the need of more research on the properties and technological characteristics of these raw materials, and promotion of their use.

### 3.3. Micronutrient Intake in Celiac Children and Adolescents

It is known that GFD can lead to imbalanced macronutrient distribution. Our previous work [26] reported that celiac children and adolescents consumed more fat and less carbohydrate than recommended and pointed at GF rendered foods as one of the culprits. Thus, taking into account directly analyzed micronutrient content, their intake on that pediatric cohort was calculated considering their age group and gender, and compared to FESNAD recommendations (Appendix A).

More than 1/4 of participants did not reach recommendations for vitamin A and vitamin E. Four out of ten children and adolescents with CD showed low intake of folic acid, which was even less than 66% of the recommendation for 25% of participants. Sixty percent of participants did not get that for vitamin D, and moreover, about 40% of them did not reach 25% of the recommendation. Most participants showed very low intakes of biotin, iodine and copper. Slightly over half the participants did not fulfil 50% of iodine recommendation and more than 40% were not able to achieve 25% of that of biotine. The intake of the rest of micronutrient was appropriate. With the exception of vitamin D, the results obtained differ from those obtained in similar pediatric research on celiac children, where low intake of iron, calcium, selenium and magnesium was observed [10,35,36].

Considering all the above mentioned, it does not seem that the GFP groups analyzed contribute to the micronutrient deficits detected in young celiac people’s diets. In fact, cereals have only a modest role as source of these micronutrients. It is important to highlight that in our previous study [26] we reported unhealthy dietary habits in these celiac children and adolescents: very low cereal and vegetable consumption, low fruit and nut intake and excessive meat consumption. Thus, general recommendations to promote healthy GFD should be given to amend the observed wrong habits. It is worth mentioning that this conclusion refers to our cohort, and that in other dietary patterns, GFPs role could be different.

It must be pointed out that, in the case of folic acid, we observed a lower content of this vitamin in GFP than in their gluten containing equivalents. In this regard, in Canada and USA [37,38] the fortification of wheat flour with folic acid is mandatory, but not for other alternative flours, such as the ones used in GFP. Taking into account the folate deficiency observed in GFD, its fortification in GFP or ingredients could be of interest for celiac children. Folate fortification measures could also be extended to biotin, whose widespread diet-deficiency in celiac population was alarming. In fact, some of the GF cereals analyzed were supplemented with this vitamin (Appendix A).

It is of interest to point out that some deficiency diseases found in celiac people, such as anemia, low bone density or zinc depletion [39] are not only justified by nutritional shortages. Other pathological situations such as systemic inflammation or intestinal microbiota alteration appear to contribute to the persistence of those deficiencies in some celiac individuals [12,40,41].

It has to be highlighted that this paper presents wide-ranging high-quality nutritional information about GF bread, pasta and cereal micronutrient content. This remains limited in the literature and even more so in food panels or in databases used for GFD design and evaluation, where it is crucial. Moreover, it has assessed not only GFP composition but also its dietetic role, discussing, in general terms, its involvement in micronutrient deficiencies of the GFD of children and adolescents. Nevertheless, extrapolation to celiac adults is limited and needs further research. Moreover, as proposed elsewhere [42], the bioavailability of GFP is a matter of concern that should also be taken into account in further studies. Finally, it is also of great interest to analyze the nutritional composition of GFPs considering their ingredients list to define the role of ingredients such as gluten free cereals or pseudocereals, starches and additives in the final composition of the product.

The practical outcomes of the present study are relevant in improving the universal guidelines for food fortification in CD [43,44]. Some individualized supplementation is usually proposed for celiac people based on micronutrient related blood monitoring. Nevertheless, GFP fortification for folate and biotin could contribute to preventing the deficiencies observed in GFD, at least in the case of celiac children and youngsters.

## 4. Conclusions

Even if lower micronutrient content was found in the analyzed GFP groups, this fact was not related with the micronutrient deficits detected in GFD in a cohort of celiac children and adolescent. Nevertheless, according to the obtained results, GFP fortification for folate and biotin seems to be a suitable proposal in order to prevent the deficiencies observed in GFD.

## Figures and Tables

**Table 1 foods-08-00321-t001:** Analytical protein and lipid content in gluten-free rendered foodstuffs divided by food groups, compared to gluten-containing products, expressed by 100 g of foodstuffs.

	Cereal Flakes			Bread			Pasta		
	GFP	GCP	*P*	GFP	GCP	*p*	GFP	GCP	*p*
Lipids	3.9 ± 5.3	2.6 ± 2.0	NS	5.6 ± 4.2	3.5 ± 4.1	0.05	3.2 ± 4.5	2.2 ± 1.1	NS
Proteins	7.4 ± 0.7	7.8 ± 3.1	NS	2.4 ± 2.0	9.0 ± 1.5	<0.001	6.5 ± 1.4	9.8 ± 4.2	NS

Values are means ± SD. SD, standard deviation; GFP, gluten-free product; GCP, gluten-containing product; *p*, statistical significance; NS, not significant.

**Table 2 foods-08-00321-t002:** Analytical [4] micronutrient content in gluten-free rendered foodstuffs divided by food groups, compared to gluten-containing products, expressed by 100 g of foodstuffs.

Micronutients/Products	Cereal Flakes					Breads					Pasta				
	GCP		GFP		*p*	GCP		GFP		*p*	GCP		GFP		*p*
	Mean	SD	Mean	SD		Mean	SD	Mean	SD		Mean	SD	Mean	SD	
Calcium (mg)	141	183	22.3	16.8	NS	60.5	34.4	90.8	57.5	NS	22.9	10.0	27.3	27.3	NS
Iron (mg)	9.87	5.19	1.8	1.8	<0.001	6.75	19.99	1.1	0.9	0.009	1.83	0.88	0.7	0.5	0.002
Sodium (mg)	332	332	357.1	313.6	NS	423	280	570.8	248.1	NS	61.0	155	34.8	65.6	NS
Zinc (mg)	5.94	12.9	0.9	0.5	NS	0.84	0.48	0.5	0.4	NS	1.45	1.10	1.1	0.5	NS
Copper (mg)	0.2	0.2	0.3	0.3	NS	0.14	0.11	0.1	0.1	NS	0.31	0.20	0.1	0.1	NS
Manganese (mg)			0.4	0.4		0.58	0.51	0.2	0.3	NS	2.22	1.18	0.5	0.8	0.042
Biotin (ug)	2.3	2.9	40.7	95.7	0.001	12.7	13.6	9.74	21.6	NS	15.8	14.8	1.10	1.00	0.002
Folate (ug)	275	35.4	55.4	88.9	0.062	32.55	14.79	32.8	56.8	NS	19.4	10.3	3.14	3.42	0.003
Niacin (mg)	16.2	8.25	3.02	2.41	<0.001	3.49	2.15	1.02	1.46	0.004	3.75	3.46	3.62	10.58	NS
Piridoxin (mg)	1.80	0.90	1.19	4.15	0.001	0.12	0.10	0.02	0.02	<0.001	0.12	0.07	0.01	0.01	<0.001
Riboflavin (mg)	1.45	0.72	0.12	0.17	<0.001	0.13	0.11	0.04	0.09	0.001	0.07	0.04	0.01	0.02	0.002
Tiamin (mg)	1.32	0.70	0.19	0.28	<0.001	0.20	0.12	0.01	0.01	<0.001	0.19	0.15	0.03	0.04	0.001
B_5_ (mg)	7.55	3.46	1.03	1.81	0.045	0.39	0.07	0.42	0.57	NS	0.70	0.40	0.29	0.36	0.067
B_12_ (ug)	0.85	0.52	3.68	4.18	0.03	0.02	0.05	88.2	236	0.026	0.04	0.08	0.73	1.37	NS
Vitamin E (mg)	3.00	6.22	0.14	0.47	0.01	0.30	0.33	1.04	1.2	NS	0.09	0.12	0.2	0.1	NS

Values are means ± SD. SD, standard deviation; GFP, gluten-free product; GCP, gluten-containing product; *p*, statistical significance; NS, not significant.

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
