# Peer review of "Micronutrient Analysis of Gluten-Free Products: Their Low Content Is Not Involved in Gluten-Free Diet Imbalance in a Cohort of Celiac Children and Adolescent"

_foods, 2019, doi:10.3390/foods8080321_

Round 1
Reviewer 1 Report
GENERAL COMENTS:
The manuscript shows current and valuable subject. This results are interesting authors analyze the average content of minerals and vitamins in many gluten-free products. An interesting complement to this work would be to show the composition of individual products or groups of products divided into the type of main raw material used or other possible division. I understand that such a supplement would involve a significant increase in the volume of work or could be the subject of the next article, so if the extension of this article is not possible, please mention it in the final conclusions. Manuscript is well organized but some proposed changes should be done. Some details in introduction, material method and discussion should be completed.
INTRODUCTION SECTION
Introduction section is rather well organized some changes could be done. There should be mentioned more others research about nutritive value of gluten-free raw material and products showed bellow.
1. Alvarez-Jubete, L., Arendt, E. K., & Gallagher, E. (2009). Nutritive value and chemical composition of pseudocereals as gluten-free ingredients. International Journal of Food Sciences and Nutrition, 60(SUPPL.4), 240–257. https://doi.org/10.1080/09637480902950597
2. Hager, A. S., Wolter, A., Jacob, F., Zannini, E., & Arendt, E. K. (2012). Nutritional properties and ultra-structure of commercial gluten free flours from different botanical sources compared to wheat flours. Journal of Cereal Science, 56(2), 239–247. https://doi.org/10.1016/j.jcs.2012.06.005
3. Matos Segura, M. E., & Rosell, C. M. (2011). Chemical Composition and Starch Digestibility of Different Gluten-free Breads. Plant Foods for Human Nutrition, 66(3), 224–230. https://doi.org/10.1007/s11130-011-0244-2
MATERIALS AND METHODS
Analytical nutrient content of GF bread, breakfast cereals and pasta
Line 62-64
It is necessary to add detailed information about the type of gluten-free products tested and their basic composition. This can be included in the text or in the form of Table.
The sentence describing information about Gluten Cereal Products included in lines 86 to 88 should be after the first sentence (lines 62-64) describing GFP. Please add in this sentence shortcut GCP.
Please add kind and type of apparatus used for Kjeldahl and Soxhlet
There is no information about method and apparatus used for moisture and ash content.
Line 66-67
The following sentence “After measuring moisture and ash content, carbohydrates were estimated by difference” should be corrected as following; After measuring protein, fat, ash content and moisture, carbohydrates .....
Lines 70-85
Please add Standards or References used for mineral and vitamin determination.
Line 81-82
Please change the order of sentences. The sentence “As a step prior to vitamin quantification, samples were extracted by liquid-liquid extraction using an aqueous acidic mixture, centrifuged and filtrated” should be at the beginning before sentence showed in line 78. It should be corrected in case if it also refers to determinations of vitamins B5 and B12.
Subject
What do you mean about the “subject” please expand the header name or add some detail at the beginning of this chapter.
Statistical analysis
I can’t find information about how many replications were done for each chemical analysis. Please completed this.
Results
Please try to discuss the results more deeply referring to the raw material composition of the tested products.
In addition try to complete discussion with references proposed to complete the introduction section.
Table 1 – In material and methods authors describe ash content measurement I don’t see this results – it should be completed.
REFERENCES
Please check abbreviations for Journals e.g. Food Chemistry should be Food Chem.
There is no indentation in positions 16 and 17
There is a mistake in positions 19 and 20. It should be completed.
There is a mistake in a 34 and 35 point of
There are errors in points 34 and 35 of References
Author Response
COMENTS:
The manuscript shows current and valuable subject. This results are interesting, authors analyze the average content of minerals and vitamins in many gluten-free products. An interesting complement to this work would be to show the composition of individual products or groups of products divided into the type of main raw material used or other possible division. I understand that such a supplement would involve a significant increase in the volume of work or could be the subject of the next article, so if the extension of this article is not possible, please mention it in the final conclusions. Manuscript is well organized but some proposed changes should be done. Some details in introduction, material method and discussion should be completed.
As mentioned by the referee, this information needs further research. Indeed, a higher number of products should be analyzed in order to classify them properly depending on their composition. Nevertheless, we agree with the referee in considering it of great interest, probably part of a future work, and we have included it in the conclusions.
Moreover, we added a Table (Supplementary Table 1S) with the ingredients of analyzed products (as appeared in the label). We can highlight that the most used ingredients for substitution of gluten are rice and corn flours and starches and that those ingredients are poorer than others (like pseudocereals) in micronutrients. Moreover, gums and mono and diglycerides of fatty acids are extensively used, sometimes hindered behind the term emulsifier, stabilizer or thickener (such as E-412 -guar gum- or E471 – mono and diglycerides of fatty acids-). Those ingredients potentially modify the nutritional composition of the product comparing to that of the gluten containing homologues.
Following referee´s suggestion, more detail on this issue has been included in the discussion section.
INTRODUCTION SECTION
Introduction section is rather well organized some changes could be done. There should be mentioned more others research about nutritive value of gluten-free raw material and products showed bellow.
Alvarez-Jubete, L., Arendt, E. K., & Gallagher, E. (2009). Nutritive value and chemical composition of pseudocereals as gluten-free ingredients. International Journal of Food Sciences and Nutrition, 60(SUPPL.4), 240–257. https://doi.org/10.1080/09637480902950597 Hager, A. S., Wolter, A., Jacob, F., Zannini, E., & Arendt, E. K. (2012). Nutritional properties and ultra-structure of commercial gluten free flours from different botanical sources compared to wheat flours. Journal of Cereal Science, 56(2), 239–247. https://doi.org/10.1016/j.jcs.2012.06.005 Matos Segura, M. E., & Rosell, C. M. (2011). Chemical Composition and Starch Digestibility of Different Gluten-free Breads. Plant Foods for Human Nutrition, 66(3), 224–230. https://doi.org/10.1007/s11130-011-0244-2Thank you very much for your contribution. Suggested references have been included in the introduction and in the discussion section, highlighting the nutritive value of some gluten-free raw materials.
MATERIALS AND METHODS
Analytical nutrient content of GF bread, breakfast cereals and pasta
Line 62-64
It is necessary to add detailed information about the type of gluten-free products tested and their basic composition. This can be included in the text or in the form of Table.
The Supplementary Table 1S has been included. It contains the tested gluten free products with information about the type of product and the ingredients declared on the label.
The sentence describing information about Gluten Cereal Products included in lines 86 to 88 should be after the first sentence (lines 62-64) describing GFP. Please add in this sentence shortcut GCP.
The sentence has been moved as suggested.
Please add kind and type of apparatus used for Kjeldahl and Soxhlet
Information has been included, as proposed by the referee.
There is no information about method and apparatus used for moisture and ash content.
We believe that is better not to include information about ash and carbohydrates in the revised manuscript, this issue has been explained to the referee in response to questions about results.
Line 66-67
The following sentence “After measuring moisture and ash content, carbohydrates were estimated by difference” should be corrected as following; After measuring protein, fat, ash content and moisture, carbohydrates .....
This part of the text has been modified.
Lines 70-85
Please add Standards or References used for mineral and vitamin determination.
Information has been included, as proposed by the referee.
Line 81-82
Please change the order of sentences. The sentence “As a step prior to vitamin quantification, samples were extracted by liquid-liquid extraction using an aqueous acidic mixture, centrifuged and filtrated” should be at the beginning before sentence showed in line 78. It should be corrected in case if it also refers to determinations of vitamins B5 and B12.
The text has been corrected as suggested.
Subject
What do you mean about the “subject” please expand the header name or add some detail at the beginning of this chapter.
The header name has been detailed as proposed.
Statistical analysis
I can’t find information about how many replications were done for each chemical analysis. Please completed this.
The asked information has been added in the text, in Materials and Methods section. Macronutrient determinations were performed in duplicate. Analysis of micronutrients was carried out once in each sample, but it was verified before the analysis that the reproducibility of the methods was less than 5%.
RESULTS
Please try to discuss the results more deeply referring to the raw material composition of the tested products.
Following reviewer’s suggestion, more detail on this issue have been included.
In addition, try to complete discussion with references proposed to complete the introduction section.
As mentioned, suggested references have been included in the introduction and in the discussion section.
Table 1 – In material and methods authors describe ash content measurement I don’t see this results – it should be completed.
We did not describe ash results because we believe that we have done a more detailed micronutrient analysis in the paper by ICP (MS, OES) and LC and that ash data do not provide more information to such analysis (moreover, ash content does not reflect the origin of the mineral material in the food product). We just used the ash content to do an approximation to carbohydrate content, which could not be directly measured. Thus, taking these considerations and reviewer`s suggestion into account, we have decided not to provide ash content and to delete carbohydrate content results from the Table 1, to describe only analytically measured macronutrients: protein and lipid content. In fact, the strength of our study is that we provide high-quality nutritional information about GFP because it is based in direct analytical results –as a gold standard for food composition determination- as suggested by Staudacher and Gibson [6].
REFERENCES
Please check abbreviations for Journals e.g. Food Chemistry should be Food Chem.
There is no indentation in positions 16 and 17
There is a mistake in positions 19 and 20. It should be completed.
There is a mistake in a 34 and 35 point of
There are errors in points 34 and 35 of References
All references has been revised and those mentioned by the referee have been corrected.

Reviewer 2 Report
This is a relevant paper, also since conclusions are based on analytically determined levels of macro- and micronutrients.
The title shows the main conclusion - and implicitly suggests that this conclusion is 'always' valid.
However, the persons involved consumed a high in meat diet (meat is rich in Zinc and a good source of other minerals ans vitamins) and presumably (as usual in Spain) with predominantly grain (wheat and maize) products based on refined (white) flour - with lower levels of micronutrients than whole wheat (and maize) products. I have serious doubts whether the conclusion stated in the title could be made when a healthier diet - lower in meat, higher in whole grain - was taken by the persons. The diets used - high meat (and, if confirmed low in wholegrain)
sHOULD be mentioned in the abstract
. Sme remarks on the dependency of the " title conclusion" and the diets consumed should also be included in the discussion.
Further remsrks
- it is a great pity that dietary fibre (often very low in GF products) is not yet included in the analysed macronutrients. Since the focus of the paper is on micronutrients, addition of fibre is no "must" but will be highly appreciated.
2.3. Dietary assessment. Can a SHORT description of the assessment method be given with reference to the full description in ref. [22].
- Page 6 " In fact, cereals are not the natural source of these micronutrients" please changge into
some ]thing such as: In fact, in the dietary intake of our group of GFpatients, cereals have only a modest role as source of these micronutrients
Page 6 - bottom replace EEUU (the Spanish abbreviation) by USA
- -Supplementary materials Table. The order (top-down) of the compounds looks somewhat strange, and may be made more logical. Most important; Chlorine. I am amazed by the big "shortage"of intake, despite the (high) salt levels. I propose to delete Chlorine from the table, it creates confusion and does not add any value.
Author Response
This is a relevant paper, also since conclusions are based on analytically determined levels of macro- and micronutrients.
The title shows the main conclusion - and implicitly suggests that this conclusion is 'always' valid.
However, the persons involved consumed a high in meat diet (meat is rich in Zinc and a good source of other minerals and vitamins) and presumably (as usual in Spain) with predominantly grain (wheat and maize) products based on refined (white) flour - with lower levels of micronutrients than whole wheat (and maize) products. I have serious doubts whether the conclusion stated in the title could be made when a healthier diet - lower in meat, higher in whole grain - was taken by the persons.
The diets used - high meat (and, if confirmed low in wholegrain) SHOULD be mentioned in the abstract
The title and the abstract have been modified in order to clarify this issue.
Some remarks on the dependency of the " title conclusion" and the diets consumed should also be included in the discussion.
More detailed information about the dietary habits of our celiac cohort has been included in the discussion section. Moreover, it has been highlighted that our conclusion refers to our cohort, and that in other dietary pattern, GFPs role could be different.
FURTHER REMARKS:
- it is a great pity that dietary fibre (often very low in GF products) is not yet included in the analysed macronutrients. Since the focus of the paper is on micronutrients, addition of fibre is no "must" but will be highly appreciated.
Thank you for your kind suggestion. We agree that fiber related data is very interesting. However, this data is part of another manuscript we hope to be published early.
2.3. Dietary assessment. Can a SHORT description of the assessment method be given with reference to the full description in ref. [22].
In fact, the paragraph that contains the mentioned sentence describes the procedure for the dietary assessment. Punctuation marks have been changed, for better understanding.
- Page 6 " In fact, cereals are not the natural source of these micronutrients" please changge into
some ]thing such as: In fact, in the dietary intake of our group of GFpatients, cereals have only a modest role as source of these micronutrients
The text has been rewritten in order to highlight referee´s suggestion.
Page 6 - bottom replace EEUU (the Spanish abbreviation) by USA
The mistake has been amended.
- -Supplementary materials Table. The order (top-down) of the compounds looks somewhat strange, and may be made more logical. Most important; Chlorine. I am amazed by the big "shortage" of intake, despite the (high) salt levels. I propose to delete Chlorine from the table, it creates confusion and does not add any value.
Thank you very much for your suggestion. Changes have been made in the final version (Supplementary Figure 1S.).
